

# Targeting super enhancers for liver disease: a review

Zhongyuan Yang, Yunhui Liu, Qiuyu Cheng and Tao Chen

Department of Infectious Disease, Tongji Hospital, Tongji Medical College Huazhong University of Science and Technology, Wuhan, Hubei, China

## ABSTRACT

**Background**. Super enhancers (SEs) refer to the ultralong regions of a gene accompanied by multiple transcription factors and cofactors and strongly drive the expression of cell-type-related genes. Recent studies have demonstrated that SEs play crucial roles in regulating gene expression related to cell cycle progression and transcription. Aberrant activation of SEs is closely related to the occurrence and development of liver disease. Liver disease, especially liver failure and hepatocellular carcinoma (HCC), constitutes a major class of diseases that seriously endanger human health. Currently, therapeutic strategies targeting SEs can dramatically prevent disease progression and improve the prognosis of animal models. The associated new approaches to the treatment of related liver disease are relatively new and need systematic elaboration.

**Objectives**. In this review, we elaborate on the features of SEs and discuss their function in liver disease. Additionally, we review their application prospects in clinical practice in the future. The article would be of interest to hepatologists, molecular biologists, clinicians, and all those concerned with targeted therapy and prognosis of liver disease.

**Methodology**. We searched three bibliographic databases (Web of Science Core Collection, Embase, PubMed) from 01/1981 to 06/2022 for peer-reviewed scientific publications focused on (1) gene treatment of liver disease; (2) current status of SE research; and (3) targeting SEs for liver disease. We included English language original studies only.

**Results**. The number of published studies considering the role of enhancers in liver disease is considerable. Since SEs were just defined in 2013, the corresponding data on SEs are scarce: approximately 50 papers found in bibliographic databases on the correlation between enhancers (or SEs) and liver disease. Remarkably, half of these papers were published in the past three years, indicating the growing interest of the scientific community in this issue. Studies have shown that treatments targeting components of SEs can improve outcomes in liver disease in animal and clinical trials.

**Conclusions**. The treatment of liver disease is facing a bottleneck, and new treatments are needed. Therapeutic regimens targeting SEs have an important role in the treatment of liver disease. However, given the off-target effect of gene therapy and the lack of clinical trials, the available experimental data are still fragmented and controversial.

Corresponding authors
Zhongyuan Yang,
1049446560@qq.com
Tao Chen, chentao_tjh@vip.sina.com

## INTRODUCTION

Enhancers that were initially identified and characterized in simian virus 40 (SV40) are referred to as a short DNA sequence in the genome (*Banerji, Rusconi & Schaffner, 1981*). As a cis-acting regulatory element, enhancers can activate gene transcription by combining with transcription factors, enhancer-binding proteins or activators (*Kadonaga, 1998*; *Ong & Corces, 2011*; *Plank & Dean, 2014*). Based on high-throughput sequencing technology, for example, chromatin immunoprecipitation sequencing (ChIP-Seq), an increasing number of enhancers have been identified. Each enhancer is critical for a subset of all the genes expressed within a specific tissue or cell type (*Levine & Tjian, 2003*; *Plank & Dean, 2014*). In recent years, a new concept has been established to define super enhancers (SEs). Several enhancers are stitched together and bound by a high number of transcription factors and cofactors (*Hnisz et al., 2013*; *Whyte et al., 2013*). SEs can activate the expression of identity-determining genes in stem cells, thereby playing a pivotal role in regulating cell fate. In addition, subsequent studies have shown that SEs are closely related to tumors, inflammation, Alzheimer's disease, atherosclerosis and autoimmune diseases (*Blinka et al., 2016*; *Brown et al., 2014*; *Chapuis et al., 2013*; *Costa-Reis & Sullivan, 2013*; *Duan et al., 2016*; *Hnisz et al., 2013*; *Hnisz et al., 2015*; *Loven et al., 2013*). To date, more effective treatment strategies in clinical practice are lacking for certain liver diseases, such as liver failure and hepatocellular carcinoma (HCC); thus, these diseases cause high mortality (*Fattovich et al., 2004*; *Moreau et al., 2013*; *Sarin et al., 2019*). There are many traditional treatments for liver disease. In recent years, due to optimized medical drug treatment, liver interventional therapy and artificial liver treatment, the quality of life and survival rate of viral hepatitis, liver cirrhosis, liver failure, liver cancer and other liver diseases have improved. It is difficult to further improve the survival rate. Liver transplantation is difficult to carry out due to the shortage of liver resources and high cost. In view of the occurrence of liver fibrosis, liver cancer and other liver diseases related to gene expression, targeted therapy for related genes is the current research focus. From the perspective of epigenetics, treating the root causes of these diseases has great prospects. Epigenetic studies have shown that SE-targeted therapy contributes to disease prognosis. Components of SEs, such as bromodomain-containing protein 4(BRD4) and CCAAT/enhancer-binding proteins (C/EBPs), have been subjected to extensive and in-depth research. Serving as a coactivator for gene transcription, BRD4 has been found to be related to liver fibrosis, HCC and hepatitis B virus infection (*Francisco et al., 2017*; *Zhang et al., 2015*; *Zhubanchaliyev et al., 2016*). C/EBP$\alpha$, an important member of the transcription factor C/EBP family, is closely associated with the aforementioned liver diseases (*Chen et al., 2000*; *Tao et al., 2012*; *Weymann et al., 2009*). Notably, pharmacologic inhibition of BRD4 or transfection of the C/EBP$\alpha$ gene has produced promising therapeutic effects in animal models (*Mei et al., 2007*; *Singh et al., 2016*; *Wang et al., 2009*; *Wang et al., 2015*). SEs appear to be promising candidates for effective therapeutic strategies against liver disease. In this article, we review a comprehensive description of SEs and recent studies that have targeted transcription factors and cofactors of SEs in liver disease and propose strategies from clinical treatment perspectives based on SEs.

## Survey methodology

We searched the Web of Science Core Collection, Embase and PubMed for peer-reviewed articles focused on (1) gene treatment of liver disease; (2) current status of SE research; and (3) targeting SEs for liver disease published from 01/1981 to 06/2022.

1. "Liver disease" was used as a basic query, and "super enhancer" or "enhancer" was added for detailed query.
2. "Liver disease" and "gene therapy" queries were used to search for information about targeted gene therapy for liver disease.

A large variety of original research articles and reviews were searched. Of the 6,783 publications retrieved from bibliographic databases for "enhancer and liver disease", 1,631 articles were classified as clinical trials or randomized controlled trials. The "gene therapy and liver disease" query returned 6,894 publications, 824 of which were clinical trials or randomized controlled trials.

At the time of writing this article, we focused on the SE components in the treatment of liver disease. Therefore, we tended to include the "super enhancer" item. A total of 121 articles describing the relationship between SE components and liver disease were investigated.

Inclusion criteria: experimental studies involving mice, rats or humans, published in English and listed in Web of Science Core Collection, Embase and PubMed starting from 01 January 1981; original experimental *in vivo* or *in vitro* studies featuring liver disease gene therapy on enhancers or SEs.

Exclusion criteria: publications before January 1st, 1981; unobtainable English version of the article. We also excluded books and documents, commentaries, summaries, editorials, and duplicate studies.

The earliest article that included SEs was published in 2015. From 2015 to 2022, a total of 50 articles met this requirement. We classified and summarized the parameters of SE exposure (structural features, biological effects) and their impact on liver disease.

## Structural and functional characteristics of enhancers and super enhancers

In contrast to promoters, enhancers are characterized by enrichment with RNA polymerase, transcription factors and coactivators (*Levine & Tjian, 2003*). Enhancers are indispensable in gene transcription. Gene transcription dramatically declines in the absence of enhancers (*Banerji, Rusconi & Schaffner, 1981*; *Shin et al., 2016*). Interestingly, enhancer-related transcription does not depend on the direction or position of the enhancer. The distance of enhancers from the target genes is variable. They can be located in the 5′ or 3′ regulatory regions, as well as within introns, and can even be located on different chromosomes (*Lomvardas et al., 2006*; *Spilianakis et al., 2005*). BRD4, C/EBPα, P300, H3K27ac, MED1, MyoD, T-bet, Oct4 + Sox2 + Nanog and Pu.1 have been used to detect the existence of typical enhancers, while some of them, such as BRD4, C/EBPα, P300, H3K27ac, MED1, MyoD and T-bet, have been used to identify SEs (*Brown et al., 2014*; *Pott & Lieb, 2015*; *Witte et al., 2015*). Currently, more than one indicator is simultaneously employed to obtain a robust set of SEs that are highly associated with cell type-specific genes (*Shin*

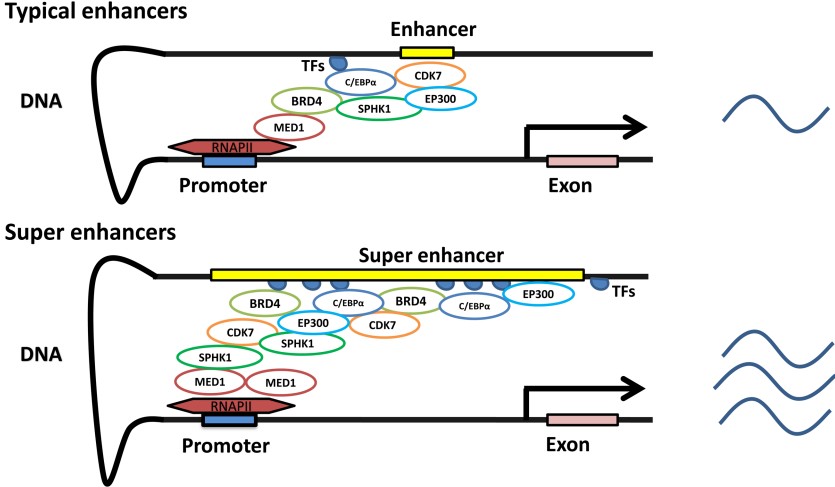

**Figure 1 Characteristics of typical and super enhancers.** Compared with typical enhancers, SEs span a larger genomic region, enriching more transcription factors, cofactors, and transcriptional activity-related histone modifications, and showing stronger transcription levels.

*et al., 2016*). SEs with unusually high enrichment with the aforementioned transcription factors and coactivators, compared to those with typical enhancers, are identified based on the development and maturation of high-throughput chromatin conformation-capturing technology. SEs can induce higher levels of target gene expression than enhancers (*Pott & Lieb, 2015*) (Fig. 1).

A three-step approach was adopted to identify the SEs (*Pott & Lieb, 2015*; *Whyte et al., 2013*). (1) Enhancers are identified according to ChIP-seq data for cell type-specific master transcription factors. (2) Constituent enhancers within 12.5 kb are stitched together to form a single large sequence. (3) The total background-normalized ChIP-seq signal for BRD4, C/EBPα, MED1, H3K27ac or other transcription factors for the stitched enhancers and the remaining individual enhancers are ranked to generate a curve, at which the slope of the plot is 1 as a cutoff to separate SEs and typical enhancers. SEs are defined as regions above the point of the curve, and the remaining enhancer regions are considered typical enhancers (Fig. 2).

SEs participate in gene transcription and determine cell fate *in vitro* and *in vivo*. In early fat synthesis in preadipocytes, hotspots that are integrated with many and various important transcription factors, such as MED1, usually constitute the core component of an SE. The differentiation of preadipocytes can be regulated by interfering with the binding of transcription factors in the hotspot region and SEs (*Siersbæk et al., 2014*). Wap, which is regulated by an SE constituted of three individual enhancers, regulates milk production in mice. Using CRISPR/Cas9 gene-editing technology, interfering with the constituent enhancers of Wap SE in the mouse genome leads to dramatically reduced expression levels of Wap genes by ~99%, indicating the important role of SE in regulating gene expression *in vivo* (*Shin et al., 2016*).

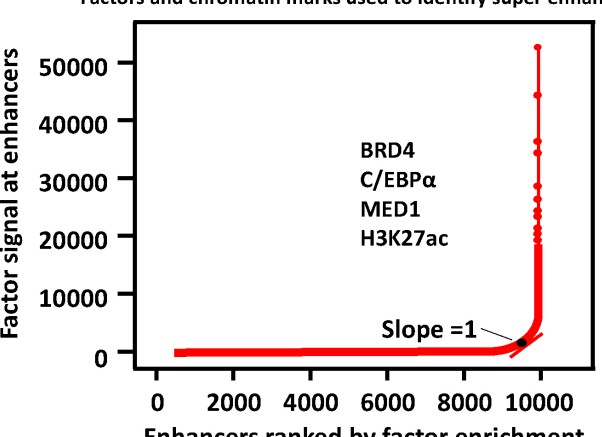

**Figure 2** **Defining super enhancers.** Typical enhancers are defined by calling peaks on ChIP-seq signal levels. All typical enhancers regions are ranked along the $x$ axis on the basis of the BRD4, C/EBP $\alpha$, MED1 or/and H3K27ac enrichment plotted on the y axis. SEs are defined as regions to the right of the inflection point of the resulting curve where the slope is one.

Disease-associated single-nucleotide polymorphisms (SNPs) are more common within SEs than typical enhancers in disease-related cells. Since it has been suggested that altered expression of cell identity genes might contribute to some diseases, intervention SEs to regulate gene expression that control and define cell identity will acquire the purpose of curing disease (*Hnisz et al., 2013*). In recent years, SEs have been found to be related to various diseases (*Shin, 2018*). Theoretically, using the SE complex as a target is more effective in patients who overexpress components of an SE complex. Based on the profound gene-regulating ability of SEs, transcription factors and cofactors have been used as novel targets in clinical research on disease therapy and show broad application prospects (*Ding et al., 2015*; *Fedorov et al., 2014*; *Tao et al., 2012*; *Zhubanchaliyev et al., 2016*).

## Super enhancers in liver disease
### Super enhancers in hepatocellular carcinoma
HCC cells exploit an aberrant transcriptional program to sustain their infinite growth and progression. Recent evidence has indicated that the continuous and robust transcription of oncogenes in cancer cells is often driven by SEs (*Loven et al., 2013*; *Sengupta & George, 2017*). SEs are also involved in driving diversified oncogenes in HCC cells. SE landscapes differ between HCC and normal liver cells, with the former being extensively reprogrammed during liver carcinogenesis. Critical components of the SE complex, namely, BRD4, cyclin-dependent kinase 7 (CDK7) and MED1, are frequently overexpressed in HCC patients and are closely associated with a poor prognosis (*Tsang et al., 2019*).

Oncogenic SEs were first identified in multiple myeloma cells, with a high assembly of BRD4 and MED1 (*Loven et al., 2013*). BRD4 is a member of the bromodomain and extra terminal domain (BET) family of proteins, which includes BRD2, BRD3, BRD4 and BRDT (*Bisgrove et al., 2007*; *Florence & Faller, 2001*; *Noguchi-Yachide, 2016*). As an important

epigenetic reader, BRD4 is the crucial component of an SE. In general, BRD4 recruits different transcriptional regulators, such as MED1 and positive transcription elongation factor b (P-TEFb), to regulate the expression of target genes (*Jang et al., 2005*; *Schnepp & Maris, 2013*; *Yang et al., 2005*). To interact with RNA polymerase II (RNAP II), P-TEFb must be recruited to transcriptional units (*Marshall et al., 1996*; *Marshall & Price, 1995*; *Peterlin & Price, 2006*). Interestingly, BRD4 is a specific activator that is known to recruit P-TEFb to transcription units (*Jang et al., 2005*; *Yang et al., 2005*). Moreover, through nonacetylated histones, such as the gamma herpesvirus f73 protein and the NF-$\kappa$B subunit RELA, BRD4 can also regulate DNA replication, the cell cycle, gene transcription and other cellular activities (*Tang et al., 2013*).

In HCC patients, *de novo* SE formation is accompanied by BRD4 redistribution (*Tsang et al., 2019*). BRD4 is significantly upregulated in HCC patients and in liver cancer cell lines, and the overexpression of BRD4 in cancer tissues is correlated with poor prognosis in HCC patients (*Tsang et al., 2019*; *Zhang et al., 2015*). In HCC patients, E2F2 was identified as the first downstream direct target of BRD4. Further experiments confirmed E2F2 as a key player in BRD4 inhibition (*Hong et al., 2016*). Another downstream target of BRD4 is Pescadillo homolog 1 (PES1), and BRD4 regulates PES1, revealing the antitumor effect of BET inhibitors in HCC (*Fan et al., 2018*). Taken together, these results shed new light on the regulation of BRD4 in HCC.

C/EBP$\alpha$ is an important member of the C/EBP family that contains a highly conserved leucine zipper domain in the C-terminus related to dimerization and DNA binding (*Osada et al., 1996*; *Ramji & Foka, 2002*; *Takiguchi, 1998*). It is known as one of the most abundant transcription factors in the liver (*Ramji & Foka, 2002*). C/EBP$\alpha$ plays a pivotal role in the control of cell proliferation and differentiation, the metabolism of phospholipids and carbohydrates, and inflammation involving hepatocytes, adipocytes and hematopoietic cells (*House et al., 2010*; *Johansen et al., 2001*; *Lekstrom-Himes & Xanthopoulos, 1998*; *Loomis et al., 2007*; *Zhao, Friedman & Fournier, 2007*). Notably, transfection of C/EBP$\alpha$ small interfering RNA (siRNA) into mice results in a 70% to 80% reduction in endogenous C/EBP$\alpha$ protein and a decrease in glycogen synthesis in the liver (*Qiao et al., 2006*; *Wang et al., 1995*). C/EBP $\alpha^{-/-}$ mice exhibit cell proliferation defects in the liver and lung (*Gery et al., 2005*). These studies show that C/EBP$\alpha$ deficiency can lead to liver metabolic disorders.

A recent study demonstrated that the SE landscape in HCC involves extensive reprogramming (*Tsang et al., 2019*). C/EBP $\alpha$ is a component in the SE complex (*Hah et al., 2015*; *Pott & Lieb, 2015*). Numerous studies have shown that C/EBP$\alpha$ is a tumor suppressor, and the expression levels of the C/EBP$\alpha$ gene and protein in normal tissues and adjacent tissues are significantly higher those that in liver cancer tissues (*Tan et al., 2005*; *Tseng et al., 2009*). Small activating RNAs (saRNAs) can upregulate CEBPA mRNA in human HCC cells and inhibit the growth of liver cancer (*Voutila et al., 2017*). The proliferation time of C/EBP$\alpha$-knockout hepatocyte cells is reduced, and atypia and tumorigenicity are increased, suggesting that C/EBP$\alpha$ promotes hepatocyte differentiation and tumor formation (*Wang et al., 2010*). In addition, downregulation of C/EBP$\alpha$ expression in hepatocellular carcinoma is significantly associated with tumor stage and shorter survival and may be used as an important marker for prognosis (*Tseng et al., 2009*). As a target

of C/EBP$\alpha$, miR-134 is downregulated in HCC cells and inhibits HCC cell invasion and metastasis (*Faraji et al., 2014*). Notably, activation of the PI3K/AKT signaling pathway in hepatoma cells blocks C/EBP$\alpha$-mediated inhibition of mitosis and proliferation, revealing that hepatoma cells might escape C/EBP $\alpha$-mediated inhibition of tumor cell proliferation *via* activation of the PI3K/AKT pathway (*Datta et al., 2007*). TNF-$\alpha$ inhibits the activity of the proximal promoter of C/EBP $\alpha$ genes, thereby decreasing the expression of C/EBP $\alpha$ mRNA and protein in the Hep3B human cell line (*Foka et al., 2009*).

Ubiquitin-proteasome system (UPS)-mediated C/EBP$\alpha$ elimination accelerates the growth of liver cancer in mice (*Wang et al., 2010*). The mechanism of C/EBP$\alpha$ elimination during carcinogenesis may be associated with elevation of gankyrin, a protein interacting with the serine 193 isoform of C/EBP$\alpha$. Metallothionein (MT) has been closely related to liver cancer (*Datta et al., 2007*). The expression of MT-1 and MT-2A in HCC is obviously decreased. Phosphorylation of the T222 and T226 sites of the C/EBP$\alpha$ gene by GSK-3 (glycogen synthase kinase 3) increases MT expression. If the downstream target gene of GSK-3 is blocked, the expression of the MT gene is significantly reduced.

CDK7 is a key gene that maintains the normal cell cycle and plays an important role in RNA transcription (*Fisher, 2005*). Previous studies have illustrated that CDK7 is highly expressed in a variety of tumors and is closely related to the progression and prognosis of these tumors. CDK7 has emerged as a therapeutic target in the treatment of tumors and inflammatory diseases (*Fisher, 2019*). As a component of the SE complex, CDK7 is frequently amplified in HCC patients (*Tsang et al., 2019*). Inhibition of CDK7-mediated expression of CDK1 or p34 Cdc2 (Cdc2) can lead to G0/G1 cell cycle arrest to exert an antitumor effect (*Bi et al., 2019*).

MED1 is a subunit of the mediator complex, a coactivator involved in regulating the transcription of RNAP II-dependent genes. It functions as a bridge to transmit information from gene-regulating proteins to RNAP II transcription machinery, serving as a scaffold for the assembly of a functional preinitiation complex with RNAP II and transcription factors (*Allen & Taatjes, 2015*; *Malik & Roeder, 2010*; *Whyte et al., 2013*). MED1 was initially used to identify SEs (*Whyte et al., 2013*). Many studies have shown that MED1 is closely related to various cancers, including HCC (*Cai et al., 2019*; *Jia, Viswakarma & Reddy, 2014*; *Leonard & Zhang, 2019*; *Tsang et al., 2019*). Liver-specific MED1 null mice do not develop liver tumors, as MED1-deficient hepatocytes do not show increased proliferation or progress to liver cancer (*Jia, Viswakarma & Reddy, 2014*; *Matsumoto et al., 2010*).

Notably, nonalcoholic fatty liver disease (NAFLD)-associated HCC shows activation of Sirtuin 7 SE (SIRT7) (*Wu et al., 2022*). In hepatoma cells, depletion of SIRT7 SE induces H3K18 acetylation and reactivation of key metabolic and immune regulators, leading to significant suppression of tumorigenicity *in vitro* and *in vivo*. Long noncoding RNA (lncRNA)-DAW is driven by a liver-specific SE and is transcriptionally activated by HNF4G, promoting liver cancer cell proliferation through activation of the Wnt/b-catenin pathway (*Liang et al., 2021*). Ajuba LIM protein (AJUBA) is an SE-associated gene regulated by transcription factor 4 (TCF4) in HCC cells (*Zhang et al., 2020*). High AJUBA expression in HCC patients is associated with an invasive phenotype and unfavorable result. Otherwise,

depletion of the AJUBA gene markedly decreases cell migration and invasion both *in vitro* and *in vivo*.

## Super enhancers in liver fibrosis

Liver fibrosis is a kind of chronic liver disease that endangers human health and is a risk factor for progression into liver cirrhosis, liver cancer and liver failure. Therefore, timely and effective treatment of liver fibrosis is very important. Recent studies have demonstrated that SEs can regulate the progression of lung and cardiac fibrosis (*Horie et al., 2018*; *Micheletti et al., 2017*). Components of an SE complex, such as BRD4, C/EBPα, CDK7 and MED1, have been subjected to in-depth research and found to be associated with liver fibrosis.

Hepatic stellate cell (HSC) proliferation and extracellular matrix secretion are the core links in liver fibrosis. As a genomic regulator, BRD4 plays a role in the response to stimulation of HSCs and directs the fibrotic response through its interaction with myofibroblast transcription (*Ding et al., 2015*; *Zhubanchaliyev et al., 2016*). The CXC chemokine CXCL6 (GCP-2) enhances the SMAD3-BRD4 interaction and directly promotes BRD4 binding to the C-MYC promoter and CMY-C binding to the EZH2 promoter *via* the SMAD2/BRD4/C-MYC/EZH2 pathway, thereby inducing the expression of profibrogenic genes in HSCs (*Cai et al., 2018*).

A significant change in HSC activation leads to the reduction or even disappearance of intracellular lipid droplets (*Kasakura et al., 2014*). C/EBPα gene expression is one of the important regulators of adipocyte differentiation. Its main function is to promote the entry of adipocytes into terminal differentiation, activating HSCs and inducing their apoptosis to inhibit the progression of liver fibrosis. In addition, C/EBPα expression can reduce liver fibrosis *in vitro* (*Tao et al., 2015*). The possible mechanism is that autophagy is involved in the expression of C/EBPα, but this mechanism needs to be further validated. Scientists have found that C/EBPα induced the apoptosis of HSCs in mice but exerted no significant effect on hepatocytes. Overexpression of C/EBPα upregulates p53 gene expression through PPAR γ and then upregulates Fas, tumor necrosis-related factor apoptosis-inducing ligand and DR5 expression, thereby inducing the apoptosis of HSCs and inhibiting liver fibrosis (*Wang et al., 2009*). Importantly, transfection of the C/EBPα gene ameliorates CCL4-induced hepatic fibrosis in mice (*Mei et al., 2007*). Therefore, C/EBPα is believed to play an important role in the pathological process of liver fibrosis.

## Super enhancers in viral hepatitis

Hepatitis B virus (HBV) infection with HBV reactivation has a high incidence and mortality (*Choi & Lim, 2017*; *Garg et al., 2011*; *Solay et al., 2018*). For example, in patients with B-cell lymphoma receiving obinutuzumab or rituximab immunochemotherapy, 8.2% showed HBV reactivation (*Kusumoto et al., 2018*). Hepatitis B reactivation can lead to more than 39.3% short-term mortality of patients with HBV-ACLF. Therefore, the mechanism underlying HBV reactivation needs to be further clarified. Recent research has shown that treatment with the BRD4 inhibitor JQ1 stimulates HBV transcription and increases the occupancy of BRD4 in the HBV genome in HepG2.2.15 cells (*Francisco et al., 2017*).

Moreover, as a catalytic subunit of the NuA4 complex, TIP60 localized to the chromatin complex of HBV cccDNA inhibits HBV replication and catalyzes histone H4 acetylation to recruit BRD4 (*Nishitsuji et al., 2018*). These results suggest that the redistribution of BRD4 is accompanied by *de novo* SEs and imply that this redistribution plays an important role in the molecular mechanism of HBV replication and reactivation, thus providing evidence for clinical therapy.

In HBV, C/EBPα at low concentrations can bind to the hepatitis B core promoter/enhancer to activate hepatitis B virus gene transcription, while C/EBPα at high concentrations can inhibit hepatitis B virus gene transcription (*Lopez-Cabrera et al., 1990*). As a multifunctional protein, p21 not only directly binds to DNA but also binds to the C/EBPα domain, activating the HBV core promoter in the form of the p21/C/EBPα complex and promoting high expression of C/EBPα (*Chen et al., 2015*). Another study finds that hepatitis B virus X protein (HBX) promotes the expression of p21, and HBX also binds to the leucine zipper domain of C/EBPα to activate transcription of the pregenomic promoter of HBV (*Park et al., 2000*). Moreover, IL-4 downregulates the expression of C/EBPα and inhibits its binding to the core upstream regulatory sequence/enhancer site, thereby inhibiting HBV DNA expression and replication (*Lin et al., 2003*).

## Super enhancers in fatty liver

An animal study provided a new mechanism for liver fat accumulation and the clinical application of JQ1 in the treatment of fatty liver. It is reported that there are some upregulated genes related to lipid accumulation. Fructose force-feeding enhances histone acetylation and BRD4 binding to the transcription region of fructose-inducible genes, thereby inducing lipid accumulation in the livers of mice. Interestingly, administration of JQ1 inhibits this process (*Yamada et al., 2016*). Moreover, induction of adipogenesis in fibroblasts leads to dynamic BRD4 redistribution to *de novo* SEs that control adipocyte differentiation. Inhibition of bromodomain-containing proteins of the BET family prevents BRD4 from assembling at these *de novo* SEs and disrupts related gene transcription, thus blocking adipogenesis (*Brown et al., 2018*).

C/EBPα plays a pivotal role in maintaining lipid homeostasis in the liver. Lipogenesis-related genes acetyl-CoA carboxylase, fatty acid synthase and sterol regulatory element-binding protein-1c are robustly suppressed by C/EBP α siRNA in db/db mice (*Qiao et al., 2006*). Expression of C/EBPα mRNA increases significantly in the NAFLD mouse model compared with mice fed a normal diet (*Hossain et al., 2018*; *Park et al., 2019*). Reduction in the expression of adipogenic protein C/EBPα ameliorates high-fat diet (HFD)-induced obesity and a related metabolic disease, hepatic steatosis (*Hu et al., 2019*; *Seo et al., 2018*). The transcription factor C/EBPα shows the greatest dominant-repressive effect on small heterodimer partner (SHP) expression in HepG2 and human hepatocytes, leading to the progression and severity of NAFLD (*Benet et al., 2015*).

MED1 is critical for the development of NAFLD in mice. MED1-knockout mice that receive a HFD for up to 4 months fail to develop fatty liver, whereas mice with MED1 that receive a HFD develop severe hepatic steatosis (*Bai et al., 2011*). MED1 may be considered a potential therapeutic target for NAFLD.

## Super enhancers in other liver diseases

The mutation in the BRD4 gene in mice produces a null allele. BRD4 heterozygotes exhibit prenatal and postnatal growth defects related to a decreased proliferation rate. In addition, mutant mice also display various anatomical abnormalities, including abnormal liver cells (*Houzelstein et al., 2002*). In alcoholic hepatitis (AH), liver neutrophil infiltration is an important pathological feature. Chemokine expression significantly increases through activated cytokine pathways. Additionally, multiple CXCL chemokines are upregulated through RNA-seq and histone modification ChIP-seq of human liver explants. Epigenome editing (for example, dCas9-KRAB) or pharmacologic inhibition of BET proteins, which are important to SE function, decrease chemokine expression *in vitro* and decrease neutrophil infiltration in murine models of AH (*Liu et al., 2021*).

## Targeting super enhancers for liver disease treatment

Recently, SEs have emerged as novel therapeutic agents in the epigenetic field (*Shin, 2018*). Drug development against components of SEs has become a hotspot. Interestingly, BRD4, C/EBPα, CKD7, MED1 and other factors have been extensively studied and some drugs have already been tested in clinical trials (Tables 1 and 2 and Fig. 3).

Using the CRISPR/Cas9 gene-editing approach, knockout of BRD4, MED1 and CDK7 significantly represses cell proliferation and colony formation in both Huh7 and HepG2 cell lines *in vitro* (*Tsang et al., 2019*), suggesting that the components of the SE complex may serve as therapeutic targets for HCC.

Based on the chemical structure, potent BET inhibitors have been classified into three classes: isoxazoles, amides/urea and 1′2′4-triazoles (*Fedorov et al., 2014*). IBET-762 and JQ1, common BRD4 inhibitors, inhibit BRD4 proteins by binding to acetylated histones (*Filippakopoulos et al., 2010*; *Wyce et al., 2013*).

The small-molecule inhibitors JQ1 and OTX015 targeting BRD4 in HCC cells substantially suppress the expression of the HCC-SE gene. JQ1 treatment also dramatically inhibits HCC-SE gene expression in HCC cell lines (*Tsang et al., 2019*). In HCC cells, JQ1 exerts an anti-proliferative effect by inducing cell cycle arrest and apoptosis (*Li et al., 2016*). Short hairpin RNA (shRNA) suppresses BRD4 expression, leading to impaired HCC cell proliferation, migration and invasion (*Li et al., 2016*; *Wang et al., 2015*). Additionally, miR-329 suppresses HCC cell invasion by regulating BRD4 expression. Upregulated expression of miR-329 results in the inhibition of BRD4 mRNA and protein expression (*Zhou et al., 2016*). As the active part in the SF1126 prodrug, LY294002 combines with BRD4 and prevents BRD4 from interacting with acetylated histone-H4 proteins on chromatin, thus replacing BRD4 coactivator proteins from the transcriptional start site of MYC in the Huh7 and SK-Hep1 HCC cell lines (*Mahadevan et al., 2012*; *Singh et al., 2016*). As a dual PI3K/BRD4 inhibitor, SF1126 has completed a phase I clinical trial in humans, showing a good safety profile, and it is hoped that it will be applied to patients in the future.

JQ1 eliminates cytokine production and blunts LPS-induced cytokine storms in mice by reducing inflammation levels, thus rescuing mice from death (*Belkina, Nikolajczyk & Denis, 2013*). Similarly, JQ1 abrogates cytokine-induced HSC activation and BRD4-mediated profibrotic enhancer activity in mice (*Ding et al., 2015*; *Zhubanchaliyev et al.,*

Peer*J*

**Table 1** Examples of small-molecule inhibitors targeting super enhancers component in liver diseases.

| Small-molecule inhibitor | Target factor | Liver disease | Target model | Clinical phase | Main results | References |
|---|---|---|---|---|---|---|
| Xylocydine | CDK1, CDK2 and CDK7 | Hepatocellular Carcinoma | Mouse (SNU-354 cells are injected subcutaneously into male athymic nude mice) | – | Xylocydine inhibits growth of HCC xenografts in Balb/C-nude mice | *Cho et al. (2010)* |
| SF1126 | PI3K/BRD4 | Hepatocellular Carcinoma | Mouse (SK-Hep1 cells or Huh-7 cells are injected subcutaneously into the NSG mouse) | Human clinical trial Phase I | Treatment with SF1126 either alone or in combination with sorafenib show significant anti-tumor activity in HCC | *Singh et al. (2016)* and *Mahadevan et al. (2012)* |
| JQ1 | BRD4 | Hepatocellular Carcinoma | SK-Hep1, Huh7 and HepG2 cells | – | BRD4 inhibition causes MYC-independent large-scale gene expression changes in liver cancer cells | *Hong et al. (2016)* |
| JQ1 | BRD4 | Hepatocellular Carcinoma | Mouse (HCCLM3 and Hep3B HCC xenograft models) | – | JQ1 inhibits tumor growth in HCC mouse model | *Li et al. (2016)* |
| THZ1 | CDK7 | Hepatocellular Carcinoma | Mouse (Luciferase-labeled Huh7 cells are injected into the liver of nude mice to establish liver tumor) | – | THZ1 significantly reduces the liver tumor size | *Tsang et al. (2019)* |
| JQ1 | BRD4 | Hepatocellular Carcinoma | HepG2 and Huh7 cells | – | JQ1 reduces the expression of the HCC-SE genes in HCC cell lines | *Tsang et al. (2019)* |
| THZ1 | CDK7 | Hepatocellular Carcinoma | HepG2 and Huh7 cells | – | SE-associated genes acquired in HCC cells is substantially reduced | *Tsang et al. (2019)* |
| SKI-II | SPHK1 | Hepatocellular Carcinoma | HepG2 and Huh7 cells | – | SKI-II strikingly abolishes the cell proliferation and colony formation of HCC cells | *Tsang et al. (2019)* |
| CBP30 | EP300 | Hepatocellular Carcinoma | HepG2 and Huh7 cells | – | CBP30 represses the expression of the 13 HCC-SE-genes | *Tsang et al. (2019)* |

Yang et al. (2023), *PeerJ*, DOI 10.7717/peerj.14780

**Table 1** (*continued*)

| Small-molecule inhibitor | Target factor | Liver disease | Target model | Clinical phase | Main results | References |
|---|---|---|---|---|---|---|
| 3,3-difluorinated tetrahydropyridinol compound | CDK7 | Hepatocellular Carcinoma | Mouse (HepG2 cells are inoculated subcutaneously into the BALB/c male nude mice) | – | 3,3-difluorinated tetrahydropyridinol compound suppresses tumor growth of HepG2 cell xenografts in nude mice | *Bi et al. (2019)* |
| MTL-CEBPA | C/EBP | Hepatocellular Carcinoma | Human (adults with advanced hepatocellular carcinoma with cirrhosis, or resulting from nonalcoholic steatohepatitis or with liver metastases) | Human clinical trial Phase I (NCT02716012) | MTL-CEBPA is the first saRNA in clinical trials and demonstrates an acceptable safety profile and potential synergistic efficacy with TKIs in HCC | *Sarker et al. (2020)* |
| JQ1 | BRD4 | Hepatocellular Carcinoma | HepG2 and LO2 cells | – | JQ1 significantly reduces SIRT7 mRNA levels | *Wu et al. (2022)* |
| JQ1 | BRD4 | Liver Fibrosis | Mouse (carbon tetrachloride-induced fibrosis in mouse) | – | JQ1 is protective and can reverse the fibrotic response | *Ding et al. (2015)* |
| JQ1 | BRD4 | Liver Fibrosis | Mouse (CCl4 induces liver fibrosis in twenty male Swiss albino mice) | – | A synergistic reduction in $\alpha$-SMA is observed when cells are co-treated with JQ1 and atorvastatin loaded NPs | *Hassan et al. (2019)* |
| JQ1 | BRD4 | Fatty Liver | Mouse (Fructose intake induces hepatic steatosis by activating fat synthesis) | – | JQ1 reduces expressions of fructose-inducible genes, histone acetylation and BRD4 binding around these genes | *Yamada et al. (2016)* |
| JQ1 | BRD4 | Fatty Liver | Adipocytes | – | JQ1 can attenuate adipogenesis | *Brown et al. (2018)* |
| iBET151 | BRD4 | Alcoholic Hepatitis | Mouse (using the NIH/NIAAA 10 days chronic-binge alcohol feeding protocol to mimic the histopathology of human AH in mice) | – | iBET151 reduces Cxcl expression and neutrophilic infiltration in murine model of AH | *Liu et al. (2021)* |

Yang et al. (2023), *PeerJ*, DOI 10.7717/peerj.14780

**Table 1** (*continued*)

| Small-molecule inhibitor | Target factor | Liver disease | Target model | Clinical phase | Main results | References |
|---|---|---|---|---|---|---|
| IL-4 | C/EBP $\alpha$ | Hepatitis B virus | Hep3B cells | – | IL-4 inhibits C/EBP $\alpha$ binding to hepatitis B core promoter/enhancer, thus inhibiting HBV DNA expression and replications | *Lin et al. (2003)* |
| JQ1 | BRD4 | Hepatitis B virus | HepG2.2.15 cells | – | JQ1 stimulates HBV transcription and increases the occupancy of BRD4 in the HBV genome | *Francisco et al. (2017)* |

**Notes.**

The table is sorted by disease type and publication time.

BRD4, bromodomain-containing protein 4; C/EBPs, CCAAT/enhancer-binding proteins; CDK7, cyclin-dependent kinase 7.

Yang et al. (2023), *PeerJ*, DOI 10.7717/peerj.14780

Peer

**Table 2** Examples of Interference tools targeting super-enhancers component in liver diseases.

| Target model | Factors for SE identification | Target gene | Interference tools | Liver disease | Main results | References |
|---|---|---|---|---|---|---|
| Mouse | C/EBP | SEs component C/EBP $\alpha$ | C/EBP-$\alpha$ gene knock-in | Hepatocellular Carcinoma (using diethyl-nitrosamine to induce hepatocellular carcinoma in our mice) | The C/EBP-$\alpha$ knock-in mice produce half the number of hepatocellular nodules | *Tan et al. (2005)* |
| Mouse | C/EBP | SEs component C/EBP $\alpha$ | C/EBP $\alpha$-S193D knock-in | Hepatocellular Carcinoma (C/EBP $\alpha$-S193D knock-in mice) | S193D mutation mice cause chromatin remodeling leading to histological appearance of 'foci-like' nodules, which are also observed in liver of old mice | *Jin et al. (2010)* |
| Mouse | MED1 | SEs component MED1 | CRISPR/Cas9 | Hepatocellular Carcinoma (diethylnitrosamine-induced hepatocarcinogenesis) | No proliferative expansion of PBP/MED1 null hepatocytes is noted in the PBP/MED1(DeltaLiv) mouse liver | *Matsumoto et al. (2010)* |
| HepG2, Hep3B, SMMC-7721, Bel-7402 and Huh7 cells | BRD4 | SEs component BRD4 | shRNA | Hepatocellular Carcinoma(HepG2, Hep3B, SMMC-7721, Bel-7402 and Huh7 cells) | shRNA can suppress BRD4 expression in HCC cells, and result in impaired HCC cell proliferation, migration and invasion | *Wang et al. (2015)* |
| Mouse | BRD4 | SEs component BRD4 | shRNA | Hepatocellular Carcinoma (The tumor cells are inoculated subcutaneously in nude mice to produce the model) | BRD4 shRNA significantly inhibits HCC cell proliferation *in vivo* | *Zhang et al. (2015)* |
| HepG2 and Hep3B cells | C/EBP $\alpha$ | SEs component C/EBP $\alpha$ | saRNA | Hepatocellular Carcinoma (HepG2 and Hep3B cells) | saRNA inhibits growth of liver cancer cell lines *in vitro* | *Voutila et al. (2017)* |
| HepG2 and Huh7 cells | BRD4, MED1, EP300, CDK7, SPHK1 | SE-complex components | CRISPR/Cas9 | Hepatocellular Carcinoma(HepG2 and Huh7 cells) | The inactivation of CDK7, SPHK1, BRD4, EP300, and MED1 attenuates cell proliferation, colony formation and the cell migratory ability | *Tsang et al. (2019)* |

Peer J

**Table 2** (*continued*)

| Target model | Factors for SE identification | Target gene | Interference tools | Liver disease | Main results | References |
|---|---|---|---|---|---|---|
| Mouse | SPHK1 | SE-complex components | CRISPR/Cas9 | Hepatocellular Carcinoma(Huh7 cells in nude mice) | knockdown of SPHK1 drastically abolishes hepatic tumor growth and pulmonary metastasis | *Tsang et al. (2019)* |
| Mouse | CDK7 | SE-complex components | CRISPR/Cas9 | Hepatocellular Carcinoma(HepG2 and Huh7 cells in nude mice) | The knockout of CDK7 in HCC cells completely abolishes the incidence of tumor formation in the nude mice | *Tsang et al. (2019)* |
| HepG2 and Huh7 cells | H3K27ac, TCF4 | SEs component TCF4 | shRNA, CRISPR/-Cas9 | Hepatocellular Carcinoma (HepG2 and Huh7 cells) | Inhibition TCF4 inhibits HCC cell migration and invasion ability | *Zhang et al. (2020)* |
| HepG2 cells | H3K27ac, HNF4G | SEs component HNF4G | shRNA | Hepatocellular Carcinoma (HepG2 cells) | Knockdown of HNF4G suppresses gene transcription of lncRNA-DAW to prevent tumor growth | *Liang et al. (2021)* |
| HepG2 and LO2 cells | H3K27ac, BRD4 | SEs component SIRT7 | CRISPR/Cas9 | Hepatocellular Carcinoma (HepG2 and LO2 cells) | Depletion of SIRT7 SE in hepatoma cells induces global H3K18 acetylation and reactivates key metabolic and immune regulators, leading to marked suppression of tumorigenicity *in vitro* | *Wu et al. (2022)* |
| Mouse | H3K27ac, BRD4 | SEs component SIRT7 | CRISPR/Cas9 | Hepatocellular Carcinoma (4-week-old female athymic nude mice are subcutaneously injected with HepG2 WT and SIRT7 SE$^{-/-}$ cells) | SIRT7 SE$^{-/-}$ cells exhibits significant inhibition of tumor growth in a xenograft model when compared with WT cells | *Wu et al. (2022)* |
| Mouse | C/EBP | SEs component C/EBP $\alpha$ | C/EBP-$\alpha$ gene | Liver Fibrosis (CCl4-induced liver fibrosis model in mice) | Exogenous C/EBP-alpha gene reduces the content of collagens and the content of hydroxyproline | *Mei et al. (2007)* |
| Mouse | C/EBP $\alpha$ | SEs component C/EBP $\alpha$ | C/EBP-$\alpha$ gene | Liver Fibrosis (CCl4-induced liver fibrosis model in mice) | Ad-C/EBP-$\alpha$ decreases extracellular matrix deposition and $\gamma$-GT levels | *Tao et al. (2012)* |
| Mouse | C/EBP $\alpha$ | SEs component C/EBP $\alpha$ | siRNA | Fatty Liver (8-wk-old wild-type and db/db mice) | Expression of lipogenesis genes is robustly suppressed in the C/EBP $\alpha$ siRNA-treated db/db mice | *Qiao et al. (2006)* |

Yang et al. (2023), *PeerJ*, DOI 10.7717/peerj.14780

**Table 2** (*continued*)

| Target model | Factors for SE identification | Target gene | Interference tools | Liver disease | Main results | References |
|---|---|---|---|---|---|---|
| Mouse (MED1 knockout mice) | MED1 | SEs component MED1 | CRISPR/Cas9 | Fatty Liver (Mice are fed with a high-fat diet for up to 4 months) | The mice fail to develop fatty liver | *Bai et al. (2011)* |
| Mouse | C/EBP $\alpha$ | SEs component C/EBP $\alpha$ | siRNA | Fatty Liver (NAFLD) | Hepatic triglyceride content, Kleiner scores and other NAFLD related index are partially reversed by C/EBP $\alpha$ siRNA | *Hu et al. (2019)* |

**Notes.**

The table is sorted by disease type and publication time.

BRD4, bromodomain-containing protein 4; C/EBPs, CCAAT/enhancer-binding proteins; CDK7, cyclin-dependent kinase 7.

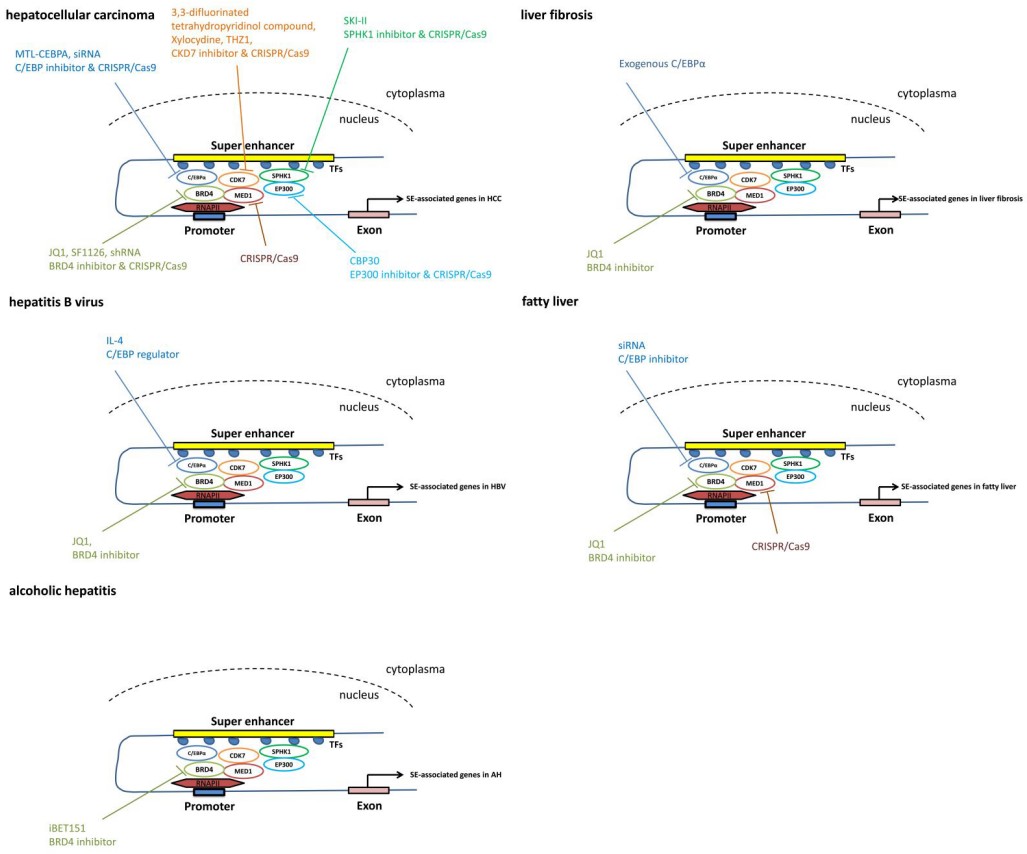

**Figure 3** **The pathways mediated *via* SEs in each individual liver disease.** Treatment and intervention targets of SEs in each individual liver disease such as liver cancer, liver fibrosis, viral hepatitis and fatty liver.

*2016*). Additionally, JQ1 reduces carbon tetrachloride (CCl4)-induced $\alpha$-smooth muscle actin ($\alpha$-SMA) expression (*Hassan et al., 2019*). These studies suggest that BRD4 of the SE complex is a potential therapeutic target for patients with liver inflammatory diseases, such as liver fibrosis and liver failure.

In a preclinical trial of liver disease, treatment with C/EBP $\alpha$ small activating RNA (saRNA) has shown significant clinical promise by reducing tumor volume and improving essential biomarkers of liver function, such as alanine transaminase, aspartate aminotransferase, bilirubin and albumin (*Reebye et al., 2014*; *Zhao et al., 2017*). This is the first saRNA therapy for a cirrhosis/HCC model to be entered in a human clinical trial.

In a recent phase I clinical trial, patients included advanced HCC with cirrhosis, resulting from liver metastases or nonalcoholic steatohepatitis. They receive MTL-CEBPA therapy, a small RNA that can upregulate C/EBP-$\alpha$. MTL-CEBPA is the first saRNA in clinical trials. This study demonstrates that MTL-CEBPA therapy is safe and has potential synergistic efficacy with tyrosine kinase inhibitors (TKIs) in HCC. These data encourage further clinical trials and combination studies of MTL-CEBPA + sorafenib in HCC (*Sarker et al., 2020*).

Xylocydine, a novel CDK7 inhibitor, can effectively induce significant cell growth inhibition and apoptosis *in vitro* and *in vivo*, making it a good candidate for HCC treatment (*Cho et al., 2010*). THZ1, another recently developed CDK7 inhibitor, can effectively suppress the growth of various cancers. It can significantly reduce liver tumor size by repressing HCC cell proliferation, demonstrating an antitumor effect (*Tsang et al., 2019*). New CDK7 inhibitors targeting the SE complex for HCC patients are worth further research.

## CONCLUSIONS

Traditional treatment methods for liver disease, including drugs, artificial liver, interventional therapy, and surgery, have made it difficult to further improve the survival rate of patients with liver disease, and liver transplantation is difficult to carry out due to the shortage of liver resources and expensive costs. We need to find new methods to further improve treatment of liver disease patients. In view of the occurrence of liver fibrosis, liver cancer and other liver diseases related to gene expression, targeted therapy for related genes is a current research hotspot. From the perspective of epigenetics, treating these diseases from the root causes has great application prospects.

The recent discovery of genome-wide SEs and the identification of their unique features in both normal development and disease progression have become hotspots in clinical and basic research. SE is characterized by a larger gene region with unusually high enrichment of binding sites for transcription factors and cofactors, which controls and defines cell identity (*Pott & Lieb, 2015*). A number of studies have suggested the key role played by SEs in many diseases and their therapeutic value in clinical applications (*Shin, 2018*). In addition, SEs can be used as promising prognostic markers in predicting disease risk and progression. Notably, we have noticed that an increasing number of studies on liver diseases and SEs are emerging. It is believed that information on SEs as therapeutic targets in liver disease will become more abundant with further research. This treatment method will change the past phenomenon of treating the symptoms but not the root causes of liver disease and solve the clinical problems from the root to achieve the goal of curing liver disease.

Super enhancer RNAs (seRNAs) are the products of SEs, which interact with various proteins and possess functional homogeneity with their host SEs (*Xiao et al., 2021*). The mechanism is reflected in three aspects: (1) regulation in cis or trans configurations; (2) promotion of chromatin looping; and (3) localization in the cytoplasm mediating various cell activities. seRNAs are mostly related to autoimmune diseases, cancers, embryonic stem cells, skeletal myoblast differentiation, and so on. Tissue specificity is an outstanding feature of SEs, particularly in the occurrence and development of embryonic stem cells and cancers. Moreover, seRNAs regulate transcription or bind to RNA-binding proteins (RBPs) to facilitate the effect of SEs.

seRNAs are involved in almost all tumor processes, including tumor formation, apoptosis, proliferation, adhesion, migration and immune response (*Tan, Li & Tang, 2020*). Cancer-related seRNAs, such as HCCL5 (*Peng et al., 2019*) and LINC01503 (*Xie et al., 2018*), play a key role in many aspects of tumor occurrence and development. In theory,

SEs can become targets of carcinogenic signals. SEs cause over-transcription of oncogenes, thus contributing to tumorigenesis. As a product of SE activation, ectopic expression of seRNAs promotes the development of cancer (*Mikhaylichenko et al., 2018*). These studies show the potential role of seRNAs as biomarkers in cancer treatment. In addition, SEs have a pivotal role in immune responses. The seRNAs IFNG-r-49 regulates the expression of IL-26 and IL-22 in inflammatory bowel disease (IBD), but does not regulate the expression of IFNG (*Aune et al., 2017*).

As a product of SEs, seRNAs may be involved in the origin of various diseases. Blocking seRNAs or SEs may be effective. Nowadays, pioneer works have promoted alternative methods to efficiently manipulate enhancer RNAs (eRNAs) levels. Many methods such as siRNA, antisense oligonucleotide (ASO), CRISPR/Cas9 and BET inhibitors can be selected for manipulating RNAs, including seRNAs. eRNA or seRNA-targeted therapy holds a great potential that is increasingly being considered (*Leveille, Melo & Agami, 2015*).

Although there is compelling evidence showing SE functions in regulating cell identity-determining genes, the degree of their involvement remains controversial. An important problem to be solved to enable epigenetic treatment of liver disease is the nonspecific nature of related drugs. For example, although BRD4 inhibitors may have shown high specificity, the BRD4 protein is involved in many gene regulatory processes and is tissue specific. Scholars can predict the toxicity of these compounds, but the mechanism of off-target effects has not been elucidated. Moreover, research in the field of liver disease is still relatively limited and mainly focuses on HCC. Hence, we urgently need more basic and clinical research to better understand the mechanisms of SEs and observe their clinical application effect to safely use them as a therapeutic target for the treatment of liver disease.

Despite various problems, epigenetic research on SEs has provided us with new therapeutic strategies for clinical practice. The roles played by individual SEs vary with downstream genes, and their overall effect in a particular liver disease, for example HCC, is relatively stable. This provides high confidence for the research of SEs in HCC. We hope more studies will provide novel directions for significant breakthroughs in the near future.

## ACKNOWLEDGEMENTS

The authors would like to thank Dr. Qin Ning for critically reading and commenting on the manuscript.

### Funding

This study was funded by the National Thirteenth ''Five Years'' Project in Science and Technology of China (2017ZX10202201, 2018ZX10302-206). The funders had no role in study design, data collection and analysis, decision to publish, or preparation of the manuscript.

### Grant Disclosures

The following grant information was disclosed by the authors:

National Thirteenth "Five Years" Project in Science and Technology of China: 2017ZX10202201, 2018ZX10302-206.

## Competing Interests

The authors declare there are no competing interests.

## Author Contributions

- Zhongyuan Yang conceived and designed the experiments, performed the experiments, analyzed the data, prepared figures and/or tables, authored or reviewed drafts of the article, and approved the final draft.
- Yunhui Liu performed the experiments, analyzed the data, authored or reviewed drafts of the article, and approved the final draft.
- Qiuyu Cheng performed the experiments, analyzed the data, authored or reviewed drafts of the article, and approved the final draft.
- Tao Chen conceived and designed the experiments, authored or reviewed drafts of the article, and approved the final draft.

## Data Availability

   This article is a literature review.

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
