# Peer review of "Targeting super enhancers for liver disease: a review"

_PeerJ, doi:10.7717/peerj.14780_

## Round 0.1 · original submission · Major Revisions

Thank you for submitting your manuscript. This article has been reviewed by relevant experts from the field. They found the manuscript carries merits, but raised various concerns which should be addressed before considering the manuscript for publication. Please refer to their comments, more specifically on the presentation of findings and binding of studies in this review.

Reviewer 1 ·

Basic reporting

The author has made a comprehensive review of the super enhancer-mediated treatment of liver diseases. The article contains valuable information, but the language has not been well polished. Some of the expressions in the manuscript are ambiguous, such as Line 175-177 “... SEs regulate gene expression that control and define cell identity… altered expression of cell identity genes might contribute to these diseases.” Line 143: “chromatins” should be “chromosomes”. Please do an overall grammar check and try to eliminate the ambiguity in the article. Otherwise, please consider getting some professional help with the language.

Experimental design

The title of the article is “Targeting super enhancers for liver disease," but the introduction mainly focuses on the definition and structure of super enhancers (SEs), very briefly mentioning the application of SEs in liver disease. It would be more appropriate to extend the introduction to indicate the significance of SEs-based treatment in liver diseases compared with all the other conventional treatments.

Please supply a proper discussion to address how the SEs fill the gap in the treatment for liver diseases in the discussion paragraph.

Validity of the findings

It would be better for the audience to capture the key information if the author could summarize and provide a featured figure to highlight the pathways mediated via SEs in each individual liver disease (hepatocellular carcinoma, liver fibrosis, viral hepatitis, fatty liver, and others).

There are different types of hepatocellular carcinoma (HCC). Please clarify and elaborate on more detailed information about SEs in different HCCs and their current available targeting treatments.

Reviewer 2 ·

Basic reporting

The topic of this review is of broad interest and within the scope of this Journal. However, related topics have been reviewed recently, see below:
1. Arechederra, María, et al. "Epigenetic biomarkers for the diagnosis and treatment of liver disease." Cancers 13.6 (2021): 1265.
2. Peng, Xiao-Fei, et al. "Targeting epigenetics and lncRNAs in liver disease: From mechanisms to therapeutics." Pharmacological Research 172 (2021): 105846.
3. Pirola, Carlos Jose, and Silvia Sookoian. "Epigenetics factors in nonalcoholic fatty liver disease." Expert Review of Gastroenterology & Hepatology 16.6 (2022): 521-536.
4. Habash, Nawras W., et al. "Epigenetics of alcohol-related liver diseases." JHEP Reports (2022): 100466.
5. Hardy, Timothy, and Derek A. Mann. "Epigenetics in liver disease: from biology to therapeutics." Gut 65.11 (2016): 1895-1905.
Although this review sought to review epigenetics in liver diseases from the aspect of super-enhancers, the messages and cited literatures are essentially still about epigenetic factors in liver diseases.

The authors only introduced the related background of super enhancers ,but did not introduce nessceary background of different liver diseases.

Experimental design

The survey methodology is okay, and sources are cited.

The review is organized into different categories of diseases. However, it lacks coherent logical transitions between different paragraphs.

Validity of the findings

Overall the goal set out in the introduction is stated and discussed. However, the discussion part lacks deep insights into the aspects of current gaps and future directions.

Additional comments

The overall writing is better to be improved and proofread. There is some inappropriate usage of words:
line 39: "raise major concerns and require dedicated evaluation"
line 42: " we systematize and summarize the features"

More importantly, I have noticed an extensive amount of identical or highly similar sentences throughout this review. Even with proper citation, the authors shall use their own words to summarize previous literature. For example:

Highly similar to other papers:
Multiple places in the survey methodology part are highly similar to this paper (Zolotova, Natalia, et al. "Harmful effects of the microplastic pollution on animal health: a literature review." _PeerJ_ 10 (2022): e13503.);
For example: from line 122 to line 131;

A large number of sentences in the session " Super enhancers in other liver diseases" are directly copied from Liu, Mengfei, et al. "Super enhancer regulation of cytokine-induced chemokine production in alcoholic hepatitis." _Nature communications_ 12.1 (2021): 1-14.
For example line 383 to 389 is almost identical to the abstract of the paper cited above.

line 282 to 287 is almost identical to sentences in Wu, Feng, et al. "Sirtuin 7 super-enhancer drives epigenomic reprogramming in hepatocarcinogenesis." _Cancer Letters_ 525 (2022): 115-130.

line 288 to 291 is also identical to sentences in Zhang, Chi, et al. "Super-enhancer-driven AJUBA is activated by TCF4 and involved in epithelial-mesenchymal transition in the progression of hepatocellular carcinoma." _Theranostics_ 10.20 (2020): 9066.

line 187 to 190 is almost identical to sentences in the abstract of Zhang, Chi, et al. "Super-enhancer-driven AJUBA is activated by TCF4 and involved in epithelial-mesenchymal transition in the progression of hepatocellular carcinoma." _Theranostics_ 10.20 (2020): 9066.

line 253 to 256 is identical to the title and some of abstract in Wang, Guo-Li, et al. "Elimination of C/EBPα through the ubiquitin-proteasome system promotes the development of liver cancer in mice." _The Journal of clinical investigation_ 120.7 (2010): 2549-2562.

line 302 to 303 is identical to the sentence in Ding, Ning, et al. "BRD4 is a novel therapeutic target for liver fibrosis." _Proceedings of the National Academy of Sciences_ 112.51 (2015): 15713-15718.

line 380 to 382 is almost identical to the sentence in Houzelstein, Denis, et al. "Growth and early postimplantation defects in mice deficient for the bromodomain-containing protein Brd4." _Molecular and cellular biology_ 22.11 (2002): 3794-3802.

line 412 to 415 is almost identical to the sentence in Singh, Alok R., et al. "Single Agent and Synergistic Activity of the “First-in-Class” Dual PI3K/BRD4 Inhibitor SF1126 with Sorafenib in Hepatocellular CarcinomaSynergistic Activity of SF1126 and Sorafenib in HCC." _Molecular cancer therapeutics_ 15.11 (2016): 2553-2562.

line 460 to 462 is identical to the sentence in He, Yi, Wenyong Long, and Qing Liu. "Targeting super-enhancers as a therapeutic strategy for cancer treatment." _Frontiers in pharmacology_ 10 (2019): 361.

Reviewer 3 ·

Basic reporting

Please make a table summarizing SEs and different liver diseases associated with them.

Please make a table summarizing different treatments of liver diseases making use of SEs.

Experimental design

The literature research method used in this manuscript is thorough. The related studies were identified and summarized.

Validity of the findings

Thank you for providing the description of the survey methodology. The method is described in detail.

Reviewer 4 ·

Basic reporting

1. Enhancers including super enhancers (SEs) primarly regulate gene expression through interacting to gene promotors. The structure of figure 1 does not show this interaction. Additionally, for RNAPII, the authors labeled it as a element on the DNA, which might be confusing to readers. I suggest the authors to revise this figure to demonstrate the interactions between promoters and enhancers. Another minor suggestion would be to modify the label of enhancers to be consistent with promoters and exons.
2. It would be great if the authors can add a column in table 2 to summurize the major findings by targetting the SEs.

Experimental design

This review mainly focused on summerizing the study of SEs at protein level, such as BRD4 and C/EBP. Emerging evidence has reported the function of some other regulators such as SE RNAs (https://pubmed.ncbi.nlm.nih.gov/34028961/). It will great if the authors can also summerized recent advances of SE RNAs, and if there is any implication of SE RNA's role in liver diseases.

Validity of the findings

The authors summerized the current theraputic strategies of targeting SE for liver diseases. However, to better help the researchers in the field, the author should not limit their survey in liver diseases. There might be other SEs related theraputic strategies applied in other disease model that haven't been investigated in liver diseases.

Additional comments

This review gave a comprehensive summerization of current stage of SEs studies in liver diseases and has its value for promoting SEs research in liver disease. However, some of the major concerns should be addressed before publication.

---

## Round 0.2 · Minor Revisions

Dear Authors, Thank you for revising the manuscript. The draft has been improved as compared to its previous version. However, there are some concerns from one reviewer that need your attention. Can you please address the reviewer`s query so a decision can be made?

Reviewer 2 ·

Basic reporting

The revised manuscript has sufficiently addressed my concern.

Experimental design

The revised manuscript has sufficiently addressed my concern.

Validity of the findings

The revised manuscript has sufficiently addressed my concern.

Reviewer 3 ·

Basic reporting

Thank you for addressing the comments. A table summarizing SEs and different liver diseases associated with them has been added to the manuscript. In addition, a table summarizing different treatments of liver diseases making use of SEs has been included as well.

Experimental design

The literature research method used in this manuscript is thorough. The related studies were identified and summarized.

Validity of the findings

Thank you for providing the description of the survey methodology. The method is described in detail.

Reviewer 4 ·

Basic reporting

no comment

Experimental design

no comment

Validity of the findings

The author mentioned that "direct manipulation of seRNAs is not yet possible", which I don't think it's true. Many methods such as siRNA, ASO, or CRISPR-Cas13 RNA editing can be used to direct manipulate RNAs, including seRNAs (https://doi.org/10.1517/14712598.2015.1029452).

Additional comments

The authors have addressed my concerns in the revision. The manuscript is suitable for publication after some minor revision.

---

## Round 0.3 · accepted · Accept

Dear Authors,

Thank you for incorporating the reviewers' suggestions.